# Physiological Cooperation between Aquaporin 5 and TRPV4

**DOI:** 10.3390/ijms231911634

**Published:** 2022-10-01

**Authors:** Kata Kira Kemény, Eszter Ducza

**Affiliations:** Department of Pharmacodynamics and Biopharmacy, Faculty of Pharmacy, University of Szeged, Eötvös u. 6, H-6720 Szeged, Hungary

**Keywords:** AQP5, TRPV4, cooperation, lung, salivary glands, uterus, adipose tissues, lens

## Abstract

Aquaporins—among them, AQP5—are responsible for transporting water across biological membranes, which is an important process in all living organisms. The transient receptor potential channel 4 (TRPV4) is a cation channel that is mostly calcium-permeable and can also be activated by osmotic stimuli. It plays a role in a number of different functions in the body, e.g., the development of bones and cartilage, and it is involved in the body’s osmoregulation, the generation of certain types of sensation (pain), and apoptosis. Our earlier studies on the uterus and the literature data aroused our interest in the physiological role of the cooperation of AQP5 and TRPV4. In this review, we focus on the co-expression and cooperation of AQP5 and TRPV4 in the lung, salivary glands, uterus, adipose tissues, and lens. Understanding the cooperation between AQP5 and TRPV4 may contribute to the development of new drug candidates and the therapy of several disorders (e.g., preterm birth, cataract, ischemia/reperfusion-induced edema, exercise- or cold-induced asthma).

## 1. Introduction

### 1.1. Aquaporins

Aquaporins (AQPs) constitute a family of small integral membrane proteins responsible for water transport across biological membranes. Water transport is essential for the survival of living cells and for many metabolic processes. Prior to the discovery of aquaporins, it had been assumed that water passes through membrane lipid barriers with passive diffusion [1,2]. Membrane water channels were discovered in 1992 by Agre et al. The first new integral membrane protein is in the human erythrocytes and is called CHIP28 (channel-forming integral protein) [3]. Soon after their discovery, it was established that these proteins function as water channels; therefore, they were designated “aquaporin, AQP”. Accordingly, CHIP28 was also renamed aquaporin-1 (AQP1) [4]. In mammals, 13 aquaporin channels (AQP0-12) have been discovered to this day. They can be divided into three groups: classical aquaporins, aquaglyceroporins, and superaquaporins [1]. There are structural and permeability differences between the groups [2]. Several studies have indicated that AQPs have important roles in the transport of various other molecules, including carbon dioxide, metalloids, nitric oxide, ammonia, urea, hydrogen peroxide, and various ions [5,6]. The functions of AQPs can be modified genetically and by phosphorylation of various amino acids, and their gating activity may also be modified depending on the intracellular and extracellular environment, including pH, oxygen pressure, temperature, and solute gradient [7]. In the human body, aquaporin channels are found in every cell [8]. 

The structure of AQPs contains six transmembrane α-helices, embedded in the cell membrane as homotetramers. Each of the four monomers contains a unique pore, functioning like a water channel, with the amino- and carboxyl-terminal regions facing toward the interior surface of the cell membrane. There are five biochemical regions (A–E) located among the α-helices that loop through the cell membrane. Two (B and E) of these regions are hydrophobic, which represent two half-helixes, containing a distinct asparagine–proline–alanine pattern (NPA motif). However, variants do exist: for example, NPC in AQP11 and NPT in AQP12, both of which are intracellular. It has been suggested that the NPA motifs are important for the targeting of aquaporins to the plasma membrane. The aromatic–arginine (ar/R) region is the other part of interest of AQPs, which consists of a strictly conserved arginine and three additional amino acids, two of which are conserved in water-specific aquaporins. This region is the narrowest point along the channel and serves as a selectivity filter, excluding anything larger than water [5]. This complex biochemical structure enables AQPs to increase water permeability through the cell membrane [9].

AQP5 belongs to the subfamily of classical aquaporins [2]. In the human body, AQP5 is found in the salivary gland, lung, gastrointestinal tract, reproductive tract, eye, and kidney [10]. This channel plays a major role in water homeostasis in these tissues. For example, AQP5 dysregulation has been involved in several diseases, including cystic fibrosis [11] and Sjörgen’s syndrome [12]. Moreover, AQP5 was found to be important in pregnancy. The presence of AQP5 is important in the mouse and rat uterus, where AQP5 was present in the apical plasma membrane of luminal epithelial cells, and there is an increase in expression in mesometrial epithelial cells at the time of implantation. In the cervices of pregnant mice, the presence of AQP3, 4, 5, and 8 was determined. AQP4 expression was low during pregnancy, and the AQP5 and 8 levels decreased at the end of pregnancy in mice. Lipopolysaccharide (LPS)-induced preterm labor had similar trends in AQP4, 5, and 8 expression in mice with normal labor at term [13]. AQP5 expression was predominant in the pregnant rat uterus, and its expression was downregulated on the day of parturition [14]. 

### 1.2. Transient Receptor Potential Vanilloid Channels

Members of the transient receptor potential (TRP) channel family are non-selective calcium-permeable proteins. The *Drosophila melanogaster* fly mutant, which was unable to respond to repeated or continuous light stimulation, was found to contain the first member of the family. The TRP family has 28 members, which are grouped into six subfamilies based on sequence homology. The six subfamilies are: melastatin (TRPM), canonical (TRPC), vanilloid (TRPV), ankyrin (TRPA), polycystic (TRPP), and mucolipin (TRPML). One of these subfamilies is the vanilloid subfamily (TRPV). The first discovered member of the TRP vanilloid subfamily (TRPV1) is activated by vanilloid compounds such as capsaicin, hence its name. The other members of the subfamily only resemble TRPV1 in their sequence; they are not sensitive to vanilloid compounds. There are six members of this receptor subfamily (TRPV1-6), all of which are non-selective calcium channels, but TRPV1-4 has a higher calcium preference than the other two members [15,16].

The TRPV4 channel was first identified in 2000 [17,18,19] and has been given a number of different names before adopting the current nomenclature, such as OTRPC4 (osmosensitive transient receptor potential channel), VR-OAC (vanilloid receptor-related osmotically activated channel), VRL-2 (vanilloid receptor-like), and TRP-12 [20].

The polymodal and non-selective TRPV4 cation channels act as a cellular mechanosensor in response to mechanical forces such as shear, stretch, osmotic swelling and shrinking, stiffening, and surface expansion, and are ubiquitously expressed in a wide range of cell types, including parenchymal cells such as smooth muscle cells, fibroblasts, epithelial cells, and endothelial cells, as well as in immune cells (macrophages, neutrophils) [21]. TRPV4 is sensitive to osmotic changes in the cell environment, and as a result of this, the activity of the TRPV4 channel can increase or decrease in a hypotonic or hypertonic solution [18]. Therefore, TRPV4 contributes to volume homeostasis both at the cellular level and systemically [18,19,20,21,22,23,24]. TRPV4 has higher permeability to Ca^2+^ and Mg^2+^ ions than Na^+^ ions and generates calcium influx when activated under normal physiological conditions. 

In general, all mammalian TRPV4 homologs are similar in length and share a high degree of sequence identity (∼95–98%). The channel protein is composed of 871 amino acids with six transmembrane (TM) domains that have both N- and C-terminal cytoplasmic tails. The pores of the channel are located in the loop between TM5 and TM6. D672 and D682 are two amino acids of TRPV4 that are key in regulating its permeability. Neutralization of both amino acids greatly reduces calcium permeability and increases the permeability of monovalent ions, suggesting that these two negatively charged groups are important for calcium binding inside the pore. D682 takes part in the ruthenium red block as well. Moreover, the phosphoinositide-binding site necessary for channel activation by physiological stimuli, hypotonicity, and heat is located in the N-terminal tail, which plays a significant role in channel regulation [20,25].

This channel is widely expressed in the smooth muscle of the cardiovascular, digestive, respiratory, and reproductive systems. Moreover, current evidence suggests that it modulates myometrial smooth muscle Ca^2+^ homeostasis in the short term to promote uterine contractility, since extracellular Ca^2+^ influx is essential for the maintenance of spontaneous, rhythmic contractions in the human myometrium, hence for parturition [25,26].

## 2. Cooperation of AQP5-TRPV4 Channels

Mammalian cells are frequently exposed to stressors causing volume changes. TRPV4 responds to isosmolar cell swelling and changes in osmolarity translated via different AQPs. Based on pharmacological logic, this phenomenon may suppose the cooperation of AQPs and TRPV4 channels. The physiological connection between TRPV4 and AQP4 channels in the brain [27] and AQP2 in renal cells is well-known [28]. AQP4 channels colocalize with TRPV4 channels in astrocytic endfeet, providing evidence for their co-participation in regulatory volume decrease. AQP2 is critical for the release of ATP induced by TRPV4 activation in the renal cortical collecting ducts cells. This ATP release occurs by an exocytic and a conductive route. Extracellular ATP (ATPe), in turn, stimulates purinergic receptors leading to ATPe-induced ATP release by a Ca^2+^ -dependent mechanism. It was proposed that AQP2, by modulating Ca^2+^ and ATP differently, could explain AQP2-increased cell migration.

We have a more modest knowledge about the cooperation of AQP5 and TRPV4. In this review, we focus on the studies in which the cooperation between TRPV4 and AQP5 channels is proved.

### 2.1. Lung

AQP5 channels are present in the alveolar epithelium of the lung, in the apical membrane of alveolar type 1 cells of mice, rats, and humans. They mediate the transport of water molecules, mostly the transcellular water pathway [29,30].

AQP5 is expressed at sites where extracellular osmolality fluctuates and exposure of mouse lung epithelial cells to hyposmolality was found to result in a dose-dependent decrease in AQP5 expression. It was found that the TRPV4 channel may have an important role in the AQP5 expression changes under hypotonic conditions. The hypotonic reduction of AQP5 was blocked by TRPV4 antagonists (ruthenium red, methanandamide, and miconazole) in lung epithelial cells.

The relationship between AQP5 expression and calcium was also examined, and it was found that, in a normal medium or calcium-containing Krebs medium diluted to generate a hypotonic medium, AQP5 expression is reduced by hypotonic stress; in contrast, nothing happens to AQP5 in the calcium-free diluted Krebs medium. This is evidence that AQP5 needs Ca^2+^ for hypotonic reduction. To prove the presumption that TRPV4 and AQP5 can cooperate, HEK (human embryonic kidney) cells were examined. When AQP5 was transfected into control HEK cells in which there was no TRPV4, AQP5 reduction was not detected, but when AQP5 was transfected into HEK cells expressing TRPV4, a significant decrease in AQP5 abundance was found. This decrease was also sensitive to ruthenium red; it could be blocked by ruthenium red. These results suggest that TRPV4 is involved in the hypotonic stress-induced modulation of AQP5 abundance [31].

Other studies also examined the relationship between TRPV4 and AQP5 in the lung. Weber et al. investigated TRPV4-/- and WT (wild type) alveolar epithelial type I (ATI) cells. It was found that the AQP5 total expression levels and plasma membrane localization are lower in the TRPV4-/- ATI cells than in the wild-type cells. This was examined with fluorescence-coupled antibodies and then supported with Western blot studies, where they found the same: AQP5 expression was reduced in the TRPV4-/- cells compared to the WT cells [32,33]. 

Sidhaye et al. examined AQP5 regulation under shear stress, and they found that AQP5 abundance was decreased as a result of shear stress, but this decrease could be prevented by the TRPV antagonist (ruthenium red). The studies were performed in primary human bronchial epithelial cells (NBHE). VGCC (voltage-gated calcium channel) was also found to participate in the regulation of AQP5. TRPV4 activation by 4αPDD (4α-phorbol 12,13-didecanoate, which is a selective TRPV4 agonist [34]) was associated with a decrease in AQP5 abundance, but this effect can be blocked by nifedipine (an agent inhibiting VGCC). In contrast, the activation of VGCC (with BayK-8644) also has the same effect on AQP5, but ruthenium red cannot prevent the decrease in AQP5 abundance [35].

There is some information about COVID-19 linkage with AQP5 and TRPV4 channels. AQP5 expression is also regulated by inflammatory cytokines, and the elevated level of TNF-α decreases AQP5 expression in mice [36], which has a high volume in inflammatory respiratory diseases such as COVID-19 [30]. The decreased AQP5 expression may induce the activation of the TRPV4 channel, which causes the loss of the alveo-capillary barrier function and subsequently the development of edema [37,38].

Based on the above, we can conclude that AQP5 and TRPV4 channels cooperate in the airways, as evidenced by several studies, but the exact mechanism of cooperation is not yet fully elucidated.

### 2.2. Salivary Gland Cells

As in the lung, signs of TRPV4 and AQP5 channel cooperation were observed in salivary gland cells. Both channels are localized in the apical region of mouse submandibular gland acinar cells [39,40]. The examination of salivary gland cells from AQP5-/- and AQP5+/+ mice revealed that when they were placed in HTS (hypotonic external solution), the HTS-stimulated Ca^2+^ entry (for which the TRPV4 channel is probably responsible) was significantly decreased in cells isolated from AQP5-/- mice [22].

To investigate the relationship between AQP5 and TRPV4 in more detail, N and C terminus-truncated AQP5 channels were formed. The expression of N terminus-deleted AQP5 suppressed TRPV4 activation and regulatory volume decrease, but not cell swelling. C-terminal deletion of AQP5 has been shown to disrupt its trafficking to the apical membrane, resulting in the retention of the protein in the cell. Furthermore, hypotonicity increased the association and surface expression of AQP5 and TRPV4. These data demonstrate that the activation of TRPV4 by hypotonicity depends on AQP5, not on cell swelling per se, and that TRPV4 and AQP5 concertedly control regulatory volume decrease [22,39,41].

### 2.3. Uterus

AQP5 and TRPV4 channels are also found in uterine tissue [26]. In our studies, we demonstrated the presence of AQP1, 2, 3, 5, 8, and 9 in the late-pregnancy rat uterus, with a prevalent accumulation of the AQP5 subtype near parturition [14]. It was established that the AQP5 expression level and myometrial contraction have an inverse correlation in the late-pregnant rat uterus [42]. AQP5 expression significantly increased during days 18–21 of pregnancy, and it dramatically decreased on the day of delivery (day 22). Additionally, it was demonstrated that oxytocin specifically downregulated this particular sort of water channel, whereas progesterone and progesterone analogs increased AQP5 expression, which was even more dominant than the estrogenic effect [14,42]. TRPV4 was reported to have potential influence on uterine smooth muscle contraction in both pregnant and non-pregnant uteri [43,44]. It was found that TRPV4 expression changes in correlation with the days of gestation, showing the lowest expression on day 18. In addition, this determined an inverse correlation in TRPV4 and AQP5 expression in the uterine tissue based on immunohistochemical studies [26]. TRPV4 expression changes through pregnancy and appears to be progesterone-dependent [45]. The progesterone analog levonorgestrel can downregulate TRPV4 expression [45], and the progesterone level falls at the end of pregnancy [46], which increases TRPV4 expression. Progesterone causes an increase in the AQP5 channel, suggesting that there be may a hormone-dependent cooperation between AQP5 and TRPV4 expression [26]. The TRPV4 channel can be activated by osmotic stimuli, and the channels are co-expressed in the uterus; therefore, AQP5 expression change triggering osmotic stress, which activates TRPV4, could be a possible mechanism for their relationship. Subsequently, TRPV4 activation causes Ca^2+^ flow, which contributes to the creation of uterine contractions (Figure 1) [26].

Citral ((2E)-3,7-dimethylocta-2,6-dienal) is a bioactive component of lemongrass, a naturally occurring TRPV4 antagonist used to investigate the relationship between TRPV4 and AQP5 channels in the late-pregnant rat uterus [47]. It was found that AQP5 expression significantly increased, and TRPV4 expression did not change significantly after citral incubation in the 22-day pregnant rat uteri, in vitro. In addition, citral treatment significantly prolonged the normal gestation period and delayed preterm delivery. 

TRPV4 and AQP5 are known to have an inverse co-expression in the late-pregnant rat uterus [26]. This may lead to the conclusion that from the 18th day of the gestation period, AQP5 expression increases, and a hypotonic milieu develops in the cytoplasm of the uterine cell, which causes smooth muscle cell membrane stretching, which triggers the activation and increased expression of the TRPV4 osmotically sensitive channel. The activation of TRPV4 leads to the Ca^2+^ current, which in turn plays a role in myometrial contractions and thus in the induction of labor. The decrease in AQP5 expression on the day of delivery can be explained as a result of cell volume regulation [47].

### 2.4. Adipose Tissues

It is a relatively new hypothesis that in obesity, where the hypothalamus receives and integrates signals from peripheral tissues (e.g., adipose and liver) and actively sends signals to manage energy balance, tissue-to-tissue co-expression (TTC) networks may reinforce communication between tissues and clarify genes or sets of genes active in one tissue that are able to induce gene activity changes in other tissues. The most connected hypothalamus gene in this network is *Aqp5*, which is linked to 169 adipose genes, while the adipose gene *Aqp5* is only linked to 2 genes in the hypothalamus. AQP5 in the hypothalamus either ‘sends’ information to the 169 adipose genes (that is, regulates the expression of the 169 adipose genes) or ‘integrates’ (responds to) their signals [48]. A study by Verkman, A. S et al. proved that AQP5-depleted mice are 10–15% smaller by weight than controls [49].

Madeira et al. made observations to reveal the presence of AQP5 in both 3T3-L1 fibroblasts and mature adipocytes by gene expression analysis. Their results show that human AQP5 is a functional water channel when expressed in adipocytes, and their role may rely on the coordination with specific functional partners in adipocytes. Moreover, the adipose tissue expresses high levels of TRPV4, and the relationship between obesity and TRPV4 has been proven in mice. Mice with a null mutation for TRPV4, or wild-type mice treated with a TRPV4 antagonist, showed elevated thermogenesis in adipose tissues and were protected from diet-induced obesity, adipose inflammation, and insulin resistance [50,51]. Based on these results, the cooperation of AQP5 and TRPV4 cannot be disregarded in adipocytes either.

### 2.5. Lens

AQP5 forms functional water channels in the rodent lens, and the dynamic membrane insertion of AQP5 may regulate water fluxes in the lens by modulating water permeability in the outer cortex [52]. A missense mutation in AQP5 (c.152 T > C, p. L51P) is associated with autosomal dominant congenital cataract, because AQP5 can participate in the maintenance of lens transparency by regulating vimentin expression via miR-124–3p.1 [53].

Lens water transport generates a hydrostatic pressure gradient that is regulated by a dual-feedback system through TRPV1 and TRPV4, to sense changes in mechanical tension and extracellular osmolarity. In this system, TRPV1 and TRPV4 reciprocally transduce changes in lens pressure into the modulation of ion transporter activity to effect alterations in circulating ion and water fluxes that act to restore the pressure gradient. 

The transport of water through the mouse lens driven by the microcirculation system can be dynamically regulated by stimuli external to the lens, too. Pilocarpine (a muscarinic agonist) treatment induces the removal of TRPV4 from the membrane, increases the hydrostatic pressure gradient, and removes AQP5 from the membranes of peripheral fiber cells located in the anterior influx pathway and equatorial efflux zone. In this situation, TRPV1/4 activation has been linked to AQP5 membrane trafficking, which increases water permeability, and the changes in zonular tension would induce localized changes in water permeability that alter the relative contributions of water influx via the anterior and posterior poles, which would in turn change the curvature of the anterior surface of the aspheric mouse lens [54,55].

### 2.6. Skin

Alterations in AQP levels have been observed in skin diseases with a defective skin barrier such as hidradenitis suppurativa, atopic dermatitis, and psoriasis. AQP5 is mainly distributed in the sweat gland secretory cells and sweat gland excretory duct cells. Downregulation of AQP5 expression was determined in hidradenitis suppurativa and atopic dermatitis [56].

AQP5 can regulate keratinocyte proliferation and differentiation, too. AQP5 overexpression in human epidermal keratinocytes (HaCaT) could induce proliferation and dedifferentiation but not influence the apoptosis of HaCaT cells, suggesting that AQP5 mediated the balance between epidermal keratinocyte proliferation and differentiation and may play an important role in maintaining the potential of keratinocytes [57]. Keratinocytes also express some members of the vanilloid subfamily of TRP channels, e.g., TRPV1, TRPV3, and TRPV4. 

TRPV4 is activated by changes in osmolarity, and it has been suggested that TRPV4 might act as a sensor of humidity or water flux from the skin surface as part of the epidermal permeability barrier homeostasis, as we have written before. TRPV4 belongs to a group of temperature-sensitive TRP channels, and it is activated by temperatures in the physiological skin temperature range. Warmth activation of TRPV4 leads to a Ca^2+^ influx through the channels, and the increase in intracellular Ca^2+^ concentration promotes Rho activation, which leads to the reorganization of the actin cytoskeleton and enhanced integrity of the intercellular junctions and, hence, enhanced barrier function. Therefore, it appears that keratinocytes might have a central role in the thermosensation of the skin that is mediated, at least in part, by TRPV4 channels [58].

AQP5 and TRPV4 proteins are both expressed in the upper layers of the palm epidermis. In an in vitro patch clamp assay, the basal activity of TRPV4 was increased in the presence of mutant AQP5 compared with the activity seen with wild-type AQP5. Therefore, AQP5 and TRPV4 might have a similar interaction in keratinocytes for detecting and responding to osmotic and mechanical stress by modulating the epidermal barrier, in particular through alterations in cell–cell junctions, such as the tight junctions [59].

Palmoplantar keratoderma Bothnia (PPKB) type is an autosomal dominant hereditary disorder, a kind of non-epidermolytic diffuse palmoplantar keratoderma. AQP5 was proven important in sweat production, possibly responsible for hyperhidrosis in patients with PPKB. A gain-of-function mutation of AQP5 results in a faster and larger increase in cell volume in hypotonic solution [60]. Moreover, the aberrant and broadened distribution of AQP5 expression in sweat glands accounts for the aquagenic wrinkling of the palms.

Cao et al. studied the AQP5Asn123Tyr variant and found that the AQP5Asn123Tyr/TRPV4 complex in keratinocytes had a stronger function than the control [60]. The co-expression of AQP5 and TRPV4 in the sweat glands suggested that the interaction might play a considerable role in the pathogenesis of hyperkeratosis and hyperhidrosis [61].

## 3. Conclusions

The activation of TRPV4 has been associated with AQP5 membrane trafficking, which influences the water permeability in several tissues. The interaction between AQP5 and TRPV4 channels is proven by the determined colocalization of these proteins in some tissues (e.g., uterus, lens, adipose tissues). On the other hand, it is well-known that the TRPV4 channels are a Ca^2+^-permeable, nonselective cation channel which is regulated by the osmotic pressure, and the AQPs have a crucial role in the regulation of the osmotic milieu. Activation of TRPV4 is induced by the exposure to hypotonicity (or intracellular hypertonicity) within the physiological range through AQP5. Based on these, we can suppose that the influence of expression through each other is relatively strong evidence for cooperation, the detailed mechanism of which can be verified with more experiments in the future.

Most information is available on their interactions in the lung, salivary gland, and uterus (Figure 2), but AQP5 and TRPV4 are suspected to interact in many parts of the body, e.g., in fat cells and the retina. Currently, pharmacological examinations of AQP5 are almost impossible because of the lack of non-toxic tissue- and subtype-selective agonists or antagonists. The regulation of AQP5 expression through TRPV4 agonists or antagonists can provide a new perspective in the study of AQP5.

We believe that the knowledge of the cooperation mechanism offers an opportunity to find new therapeutic targets, e.g., in the treatment of asthma, COVID-19 lung disease, or the prevention of premature birth.

## Figures and Tables

**Figure 1 ijms-23-11634-f001:**
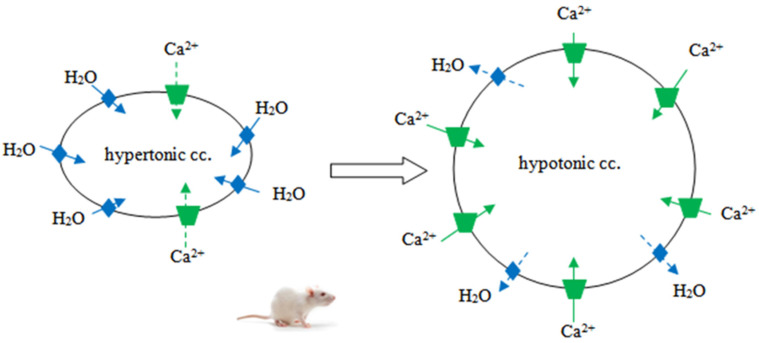
Co-expression and cooperation of TRPV4 and AQP5 influence the contractions in the late-pregnant uterus [26,47]. The osmotic stress (through the AQP5) activated the TRPV4, and TRPV4 activation triggers Ca^2+^ flow, which contributes to the creation of uterine contractions. cc.: intracellular concentration, AQP5: 
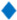
, TRPV4: 
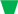
.

**Figure 2 ijms-23-11634-f002:**
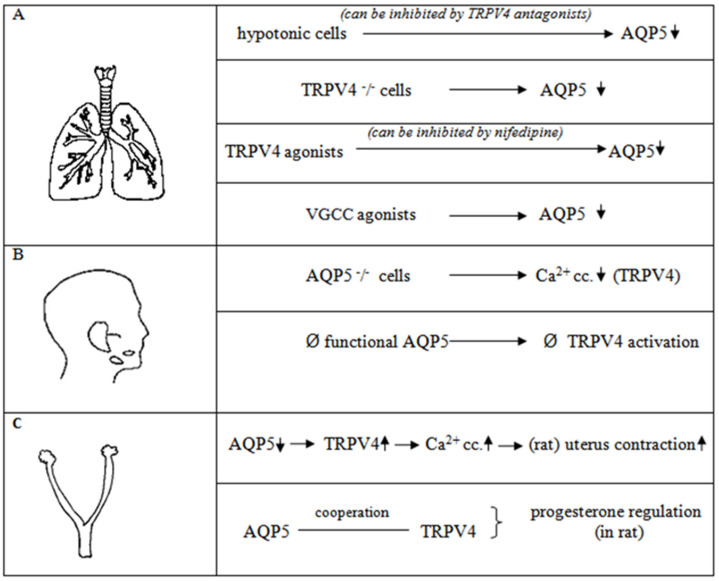
Physiological cooperation between AQP5 and TRPV4 channels in the human lung (**A**), salivary gland (**B**), and rat uterus cells (**C**). VGCC: voltage-gated calcium channel, ↓ decreased or ↑ increased expression.

## Data Availability

Research data are available for request.

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
