# Peer review of "Physiological Cooperation between Aquaporin 5 and TRPV4"

_ijms, 2022, doi:10.3390/ijms231911634_

Round 1

Reviewer 1 Report

The present review focuses on the functional cooperation between human aquaporin 5 and the TRPV4 and its role in physiology. Not very many papers deal with this cooperation, so the research field dealing with this specific aquaporin interactions is quite small. However, the cooperation seems to be important in a number of tissues. 

Aquaporin 5 is known to allow flux of other substances, like for instance hydrogen peroxide. So I wonder how does the flux of hydrogen peroxide relate to TRPV4 activity?

The paper focuses on the cooperation between AQP5 and TRPV4 under certain circumstances. One thing that is not really discussed in the paper is whether there is a direct interaction between AQP5 and TRPV4. I believe the authors should address this very important point in their review. 

I have a number of specific suggestions that appear from the attached file.

Author Response

Answers to Reviewer #1

The authors would like to express their thanks to Reviewer #1 for the questions, which have promoted the creation of a better manuscript as concerns its scientific value. Our answers are given below.

The present review focuses on the functional cooperation between human aquaporin 5 and the TRPV4 and its role in physiology. Not very many papers deal with this cooperation, so the research field dealing with this specific aquaporin interactions is quite small. However, the cooperation seems to be important in a number of tissues. 

Aquaporin 5 is known to allow flux of other substances, like for instance hydrogen peroxide. So I wonder how does the flux of hydrogen peroxide relate to TRPV4 activity?

AQP5 can also permeate hydrogen peroxide (H2O2) (Rat Aquaporin-5 Is pH-Gated Induced by Phosphorylation and Is Implicated in Oxidative Stress., Rodrigues C, Mósca AF, Martins AP, Nobre T, Prista C, Antunes F, Cipak Gasparovic A, Soveral G., Int J Mol Sci. 2016 Dec 13;17(12):2090.; Human Aquaporin-5 Facilitates Hydrogen Peroxide Permeation Affecting Adaption to Oxidative Stress and Cancer Cell Migration., Rodrigues C, Pimpão C, Mósca AF, Coxixo AS, Lopes D, da Silva IV, Pedersen PA, Antunes F, Soveral G., Cancers (Basel). 2019 Jul 3;11(7):932.), although there is no evidence that the TRPV4 activity can be influenced by endogenous H2O2. Suresh K, et al. demonstrated that the exogenous H2O2 could increase intracellular Ca2+ levels in mouse and human lung microvascular endothelial cells, which process takes place through the activation of the TRPV4 channel (Suresh K, Servinsky L, Reyes J, Baksh S, Undem C, Caterina M, Pearse DB, Shimoda LA., Hydrogen peroxide-induced calcium influx in lung microvascular endothelial cells involves TRPV4., Am J Physiol Lung Cell Mol Physiol. 2015 Dec 15;309(12): L1467-77).

The paper focuses on the cooperation between AQP5 and TRPV4 under certain circumstances. One thing that is not discussed in the paper is whether there is a direct interaction between AQP5 and TRPV4. I believe the authors should address this very important point in their review. 

The interaction between AQP5 and TRPV4 channels is proved by the determined colocalization of these proteins in some tissues (e.g. uterus, lens, adipose tissues). On the other hand, it is well known, that the TRPV4 channels are a Ca2+-permeable, nonselective cation channel which is regulated by the osmotic pressure and the AQPs have a crucial role in the regulation of the osmotic milieu. Activation of TRPV4 is induced by exposure to hypotonicity (or intra-cellular hypertonicity) within the physiological range through AQP5. 

The same phenomenon is supported by the cooperation between AQP4 and TRPV4 in astrocytes (Valentina Benfenati, Marco Caprini, Melania Dovizioso, Maria N Mylonakou, Stefano Ferroni, Ole P Ottersen, Mahmood Amiry-Moghaddam. An aquaporin-4/transient receptor potential vanilloid 4 (AQP4/TRPV4) complex is essential for cell-volume control in astrocytes Proc Natl Acad Sci USA, 2011 Feb 8;108(6):2563-8.), or in edema of meningiomas (Evaluation of AQP4/TRPV4 Channel Co-expression, Microvessel Density, and its Association with Peritumoral Brain Edema in Intracranial Meningiomas. Faropoulos K, Polia A, Tsakona C, Pitaraki E, Moutafidi A, Gatzounis G, Assimakopoulou M. J Mol Neurosci. 2021 Sep;71(9):1786-1795). The above can be considered as direct evidence of the physiological relationship between AQP5 and TRPV4 channels.

I have several specific suggestions that appear in the attached file.

We corrected the manuscript on specific suggestions of the Reviewer and marked it (and all changes in the text, too) with yellow color. The Reviewer suggested a figure about “the organization of aquaporins relative to the membrane and showing the various parts described in the text”. Unfortunately, we do not have the IT and software background to be able to create such a picture, but we have supplemented the text with additional information about the structure of aquaporin channels.

Reviewer 2 Report

 In this review Kemény and Ducza focus on the physiological cooperation between AQP5 and TRPV4 and they proposed that the better understanding of AQP5 and TRPV4 interaction may contribute to the development of new drug candidates and the therapy of several disorders. I found this review very interesting however I have minor comments:

1-     In 1.2: Transient receptor potential vanilloid channels the authors stated:

TRPV4 is sensitive to osmotic changes in the cell environment, and as a result of this, the activity of the TRPV4 channel can decrease or increase in a hypotonic or hypertonic solution (13).

The previous sentence is misleading because the activity of TRPV4 increases in hypotonic solution. Thus, I think that is better to rewrite it in the following terms:

TRPV4 is sensitive to osmotic changes in the cell environment, and as a result of this, the activity of the TRPV4 channel can increase or decrease in a hypotonic or hypertonic solution (13).

2-     In 2: Cooperation of AQP5-TRPV4 channels. The authors stated:

“The physiological connection between TRPV4 and AQP4 channels in the brain, lung, and renal cells is well known (20).“

In renal cells TRPV4 is known to be physiologically connected to AQP2 not to AQP4.

3-     In 2.1 Lung. The authors stated:

“AQP5 is expressed at sites where extracellular osmolality fluctuates, and exposure of mouse lung epithelial cells to hypotension was found to result in a dose-dependent de[1]crease in AQP5 expression”

Is hypotension or hyposmolarity?

4-     In 2.3 Uterus. Figure 1 is not very clear, what is hypertonic cc. and hypotonic cc.? Is it extracellular or intracellular hypo or hypertonicity? Or does it refer to hypertonic and hypotonic contraction?

5-     Two arrows of the legend of Figure 2 are misplaced.

6-     In Conclusions. The authors stated:

“The activation of TRPV4 has been associated with AQP5 membrane trafficking, which increases water permeability in several tissues.”

I found that this sentence could be interpreted as that TRPV4 activity increases water permeability, however, as it was previously discussed, TRPV4 activity decreased AQP5 at the cell membrane, thus water permeability decreased.

Author Response

Answers to Reviewer #2

The authors would like to express their thanks to Reviewer #2 for the questions, which have promoted the creation of a better manuscript as concerns its scientific value. Our answers are given below.

 In this review Kemény and Ducza focus on the physiological cooperation between AQP5 and TRPV4 and they proposed that the better understanding of AQP5 and TRPV4 interaction may contribute to the development of new drug candidates and the therapy of several disorders. I found this review very interesting however I have minor comments:

  • In 1.2: Transient receptor potential vanilloid channels the authors stated:

“TRPV4 is sensitive to osmotic changes in the cell environment, and as a result of this, the activity of the TRPV4 channel can decrease or increase in a hypotonic or hypertonic solution (13).” 

The previous sentence is misleading because the activity of TRPV4 increases in hypotonic solution. Thus, I think that is better to rewrite it in the following terms:

TRPV4 is sensitive to osmotic changes in the cell environment, and as a result of this, the activity of the TRPV4 channel can increase or decrease in a hypotonic or hypertonic solution (13).

 Thanks for the recommendation of the Reviewer, we change this sentence.

2-     In 2: Cooperation of AQP5-TRPV4 channels. The authors stated:

“The physiological connection between TRPV4 and AQP4 channels in the brain, lung, and renal cells is well known (20).“

 In renal cells TRPV4 is known to be physiologically connected to AQP2 not to AQP4.

 We have corrected this in the revised manuscript.

3-     In 2.1 Lung. The authors stated:

“AQP5 is expressed at sites where extracellular osmolality fluctuates, and exposure of mouse lung epithelial cells to hypotension was found to result in a dose-dependent decrease in AQP5 expression”

Is hypotension or hyposmolarity?

It is an interesting question because more literature uses hypotension as a synonym for hyposmolarity. In this sentence, the hyposmolarity can be better understood.

4-     In 2.3 Uterus. Figure 1 is not very clear, what is hypertonic cc. and hypotonic cc.? Is it extracellular or intracellular hypo or hypertonicity? Or does it refer to hypertonic and hypotonic contraction?

      The hypertonic cc. and hypotonic cc. mean the intracellular concentration in Figure 1.

5-     Two arrows of the legend of Figure 2 are misplaced.

 We have corrected this in the revised manuscript.

6-     In Conclusions. The authors stated:

“The activation of TRPV4 has been associated with AQP5 membrane trafficking, which increases water permeability in several tissues.”

I found that this sentence could be interpreted as that TRPV4 activity increases water permeability, however, as it was previously discussed, TRPV4 activity decreased AQP5 at the cell membrane, thus water permeability decreased.

We have corrected this sentence as below:

“The activation of TRPV4 has been associated with AQP5 membrane trafficking, which influences the water permeability in several tissues.”

Round 2

Reviewer 1 Report

I have included my comments in the PDF of the manuscript.

I only have one further comment. I still believe the authors should be careful when stating that the two proteins interact physically, to the best of my knowledge this has never been shown directly (co-purification, co-IP etc). Co-localization by confocal microscopy is not proof that the two proteins interact physically. However, there is evidence that they interact functionally. So I believe this show appear from the manuscript.

Author Response

Answers to Reviewer #1

The authors would like to express their thanks to Reviewer #1 for the questions. Our answers are given below.

I have included my comments in the PDF of the manuscript.

-We corrected the manuscript on specific suggestions of the Reviewer and marked it with light blue color.

I only have one further comment. I still believe the authors should be careful when stating that the two proteins interact physically, to the best of my knowledge this has never been shown directly (co-purification, co-IP etc). Co-localization by confocal microscopy is not proof that the two proteins interact physically. However, there is evidence that they interact functionally. So I believe this show appear from the manuscript.

- We agree with the Reviewer that co-localization doesn’t mean physical cooperation. The co-localization can be only one of more pieces of evidence that must be fulfilled. It has been known, that the TRPV4 is osmosensitve cation channel, and the AQPs influence the osmotic milieu, so the cooperation can be hypothesized. This was proved by the decrease of AQP5 expression in TPRV4-/- cells or after the TRPV4 agonist treatment decreased the AQP5 expression in the lung (Figure 2.). Based on these, we can suppose, that the influence of expression through each other is relatively strong evidence for cooperation, the detailed mechanism of which can be verified with more experiments in the future.